# Protest and Apology in the Arctic: Enacting Citizenship in Two Recent Swedish Films

**Lydia Kokkola** [1,2,*], **Annbritt Palo** [3] and **Lena Manderstedt** [3]

1   English & Education, Luleå University of Technology, Luleå 97187, Sweden
2   Faculty of Humanities, The University of Turku, 20014 Turku, Finland
3   Swedish & Education, Luleå University of Technology, Luleå 97187, Sweden; Annbritt.Palo@ltu.se (A.P.);
    Lena.Manderstedt@ltu.se (L.M.)
*   Correspondence: lydia.kokkola@ltu.se; Tel.: +46-(0)920-493045

**Abstract:** Today, Sweden enjoys a positive international reputation for its commitment to human rights issues, for instance, in relation to the recent migrant crisis. Abuses committed by the Swedish state against certain ethnic groups within the country are less well known, both within and beyond its borders. These included systematic attempts to curtail the use of indigenous and local languages, thereby causing communicative and ideological rifts between children and their parents. These policies were enacted through the school system from the 1920s until the 1970s, and particularly affected people living in the Arctic region where the national borders are disputed. In this article, we examine two twenty-first-century films set during this era, featuring feisty female characters responding to the school policy. *Elina: As though I wasn't there* is a children's film created by people "outside" the cultural group represented; and *Sámi Blood* features an adolescent protagonist (and her older self), created by "insiders" of the cultural group represented. In both films, the female protagonists' relative lack of agency within the state school system is contrasted with their powerful connections to the Arctic landscape. We seek to examine how these films contribute to the work of apology, beginning with a public acknowledgement of the wrongs of the past. Whilst one of the films concludes with a celebration of the female protagonists' agency, the other proffers a more ambiguous portrayal of power in relation to culture, nationality, and identity.

**Keywords:** Swedish Arctic; national minorities; school and language policy; female agency; belonging; shame and apology; contemporary films

## 1. Introduction

In 1957, the Swedish parliament passed a law preventing teachers from punishing school children who spoke their home language by either depriving them of food or inflicting corporal punishment (Kenttä and Weinz 1968). Other forms of punishment, such as verbal humiliation or exclusion continued. These punishments were part of a language policy curtailing the use of indigenous and local languages that began in the 1920s, which primarily affected people living in the Arctic region. The policy caused communicative and ideological rifts within families. Today, Sweden is so internationally renowned for its commitment to human rights issues that many are unaware of the abuses that Swedish institutions have inflicted on their own ethnic minorities in the recent past. This article examines two recent Swedish films that attempt to redress that balance by drawing public attention towards these educational practices and their impact on the community as a whole, but particularly on the young female protagonists. By comparing two different films, featuring characters from different ethnic groups, we do not intend to conflate the Sámi and Tornedalingar, even though proximity and inter-marriage have created webs of shared heritage. Rather we seek

to focus on how these films contribute to the work of apology as they portray female characters whose self-autonomy is threatened by the Swedish state's policies enacted through the school system. Acknowledging this history is a step towards reconciliation.

The two films—*Elina-som om jag inte fanns* (Elina: As if I wasn't there; Härö 2003), and *Sameblod* (Sámi Blood; Kernell 2016)—depict members from two of Sweden's five national minorities: the Tornedalingar and the Sámi. *Elina* features a girl protagonist and was directed by Klas Härö; but neither Härö, the actors, nor the author of the book upon which the story is based are Tornedalingar. In contrast, *Sámi Blood*, featuring an adolescent protagonist (and her older self), was scripted, directed, and acted by people with Sámi heritage. 'National minorities' are Swedish citizens who self-identify as belonging to an ethnic group that is deemed to be distinct from the majority of the Swedish population in terms of culture, language, religion, and/or other social practices. The group must have contributed to Sweden's culture and society for at least 100 years (i.e., three generations). The five recognized groups are the Jews, the Romani, the Sámi (the indigenous people of the European Arctic, formerly known by the pejorative 'Lapp'), Finno-Swedes, and the Tornedalingar. The latter are the least well known beyond Sweden's borders. 'Tornedalingar' literally translates as 'the people of the Torne valley'[1]. The Torne River marks the border between Finland and Sweden. This border did not exist until 1809, when Sweden ceded Finland (then a Swedish colony) to Russia as part of the peace treaty. The border remains problematic for both groups. Cultural practices, such as reindeer husbandry and fishing, have meant that many Sámi have lived a semi-nomadic life, travelling throughout Sápmi (usually referred to in English as 'Lapland'). Today, Swedish, Finnish, Norwegian, and Russian borders run through Sápmi, creating challenges for those who regard Sápmi as their home. The protection of Finnish within the National Minorities Act (SFS 2009:724) acknowledges this history and the impact on those living in the Arctic region where daily contact created a Creole language—Meänkieli—which literally translates as 'our language'. Meänkieli was not officially recognized as a language in Sweden until 2002. Prior to that it was simply referred to as 'Tornedalsfinska' (Finnish of the Torne Valley). Finnish speakers tend to regard Meänkieli as a dialect, and it has no legal status on the Finnish side of the border.

*Elina* is a children's film set in the region in 1952, however, it was created using actors from outside the region. For the most part, the actors speak standard Finnish with just a few Meänkieli words and pronunciations. The eight-year-old protagonist, Elina, primarily speaks standard Finnish as well as Swedish. At the end of the film, we learn that her father was not from the region, which also explains why her home language is closer to standard Finnish than Meänkieli. The other film we discuss was directed by a Sámi woman using Sámi actors; it is intended for an adult audience. *Sámi Blood*, set in the 1930s with a frame story set in the late 20th century, depicts adolescent Elle-Marja, who leaves Sápmi as a teenager and does not return until she attends her sister's funeral.

Both films depict national minority girl characters attending a school where the use of their ethnic language is forbidden. In *Sámi Blood*, Elle-Marja attends a residential school (known in Swedish as a 'workhouse') with her younger sister, Njenna. Their father is dead, and their mother is engaged in reindeer husbandry. At school, Elle-Marja is a good student who would like to become a teacher. Her dreams are scoffed at by both her teacher and the Sámi themselves, so she runs away, burns her Sámi dress, and calls herself Christina, thereby denying her cultural heritage and her Sámi identity. The first ten minutes of the film show Elle-Marja as an old woman, returning to Sápmi to attend Njenna's funeral. She dissociates herself from all things Sámi: the language, the culture, even her family. In the closing scenes, she enters the church to ask forgiveness from her dead sister in their language, and then walks towards the other Sámi. In *Elina*, Elina lives with her mother and sisters.

---

[1]　The official English translation of the Swedish term 'Tornedalingar' is 'Torne Valley Finns': a term which suggests they are Finnish nationals, although they are Swedes. Since many Finns live in the Torne valley, we find this term unhelpful and use the term 'Tornedalingar' untranslated. This is still awkward, but does not lead to confusion about national affiliation.

The film concentrates on the family's poverty, Elina's grief over the loss of her father and, above all, Elina's struggle for the right to speak her native language in school.

Both Elle-Marja and Elina are proficient Swedish speakers, but they prefer to speak their ethnic language with their younger sisters. The extensive use of the Arctic landscape in both films locates both girls as belonging to this physical environment: they are both able to read the cues proffered by the land to navigate safely. In relation to the films, the key difference is that Elle-Marja's mother is a reindeer herder, a semi-nomadic occupation which means her children must attend a residential school. Elina's family has a few cows and a small amount of land for growing potatoes: she is able to live at home with her mother and sisters.

Our aim is to examine how these films contribute to the work of apology as they portray female characters who demonstrate agency through their knowledge of the Arctic landscape and their resistance to school policy. We begin by outlining how the films portray the Swedish state's attempt to undermine minority cultures within the Arctic region through the school system as a form of citizenship education. We then consider how the films function in the historical context of their production: one six years before the National Minorities Act was passed (*Elina*), the other seven years after it was passed (*Sámi Blood*). Finally, we conclude by considering how the films reflect a change in citizenship education.

The National Minorities Act grants additional rights to members of the five ethnic groups, including the right to inform other Swedish citizens about the abuses of the past, the right to develop their culture, and the right to use their ethnic languages in dealings with the Swedish authorities. In 2018, a government bill (Inquiry 2017/18:199) was presented to the Parliament and was enacted into law on 1 January 2019, further protecting the rights of national minorities. The positive discrimination enshrined in these Acts is not to everyone's taste, and "street-level bureaucrats" as they are referred to by Lipsky (1980)—social workers, teachers, police, health workers, and other public employees—are able to enact or undermine the spirit of law. These are the people who transform laws into the experiences of those affected by the law, which for Lipsky means that they are, in practice, more powerful than politicians.

## 2. Schools for Suppressing Minority Languages

In 1922, Uppsala University opened the world's first state institute for racial biology: Statens institut för rasbiologi (SIFR). The Institute's first director, Herman Lundborg (1868–1943), was fascinated by the peoples of the north, especially the Sámi, and documented their racial features. This involved forcing people to be photographed naked and taking measurements of their facial features. This so-called 'research' formed the base from which policies, including forced sterilization and the suppression of minority languages, were developed. *Sámi Blood* includes a scene in which the protagonist is forced to strip in front of her teacher, classmates, and a group of 'researchers', and is measured and photographed like an object, whilst a gang of local Swedish youngsters gapes through the window. This is both a physical and psychological assault, symbolized by the sound of the flash, which hits Elle-Marja like a whip. Shots of her startled eyes are spliced with shots of the gawping boys, highlighting the sexual nature of this violation. *Elina* does not include any explicit reference to the work of SIFR, but it depicts institutionalized racism and the enactment of the Swedish-only policy, which continued for decades after the SIFR had abandoned racial purity as a topic in favor of research into medical genetics.

Workhouses were established throughout the northernmost parts of Sweden in the first years of the twentieth century. They were initially a humane response to the extreme poverty of the region, but developed into sites where Swedish-only policies were enacted. The children came from impoverished backgrounds and received free board and lodging. Benign as this sounds, the children were only allowed to use Swedish for extended periods of time, which distanced them from their families upon their return. Unlike Elle-Marja, Elina attends day school, and so can speak her home language with her family daily. The film begins with a kindly, Swedish-speaking doctor asking whether

Elina, who has been ill, would like to return to school and eat school lunches every day. The reason why this is considered such a treat becomes clear as the film progresses: Elina's mother is too poor to feed her daughters well. However, Elina's strong sense of moral justice and her determination to help her friend Anton, who only speaks Finnish, lead her into conflict with the head teacher. The battle is played out through refusing food, despite its scarcity.

The original legislation governing the establishment of workhouses in 1913 made no reference to local languages. The situation began to change in the mid-1930s, shortly before the period in which the bulk of *Sámi Blood* is set. In 1923, the instructions from the Ministry of Education were that "staff in the workhouses within Finnish speaking areas must always use Swedish with the children in everyday conversation, also during breaks" (Elenius 2006; our translation). One year later, the policy had tightened to "within Finnish speaking areas, the use of Swedish as the language of the workhouses must be ensured", which effectively heralded in the use of punishments; the same rules applied to the Sámi languages (Elenius 2006). The tightening of the policy can be gleaned from the films. In *Sámi Blood*, Elle-Marja belongs to the first generation of children to go through this system. She is a good student, and her enchantment with reading is conveyed in several key scenes. Her school teacher offers her a collection of poetry, and asks her to read out a poem by Edith Södergran: Landet som icke är (I long for the land that is not). The poem allows Elle-Marja to voice her longing for a world that is different, but despite her evident proficiency and poetic sensitivity, her teacher informs her that Sámi brains are too weak to cope with advanced education. The teacher is not unkind; she clearly believes the SIFR research that supports this belief and cannot see the contrary evidence—Elle-Marja's evident capacity to learn—standing in front of her. The belief that "Lapp ska vara lapp" ("a Lapp will be a Lapp", a phrase coined in 1906 by Vitalis Karnell who, at the time, was the vicar of Sweden's northernmost municipality, Karesuando) also permeates the Sámi society visible in the film: Elle-Marja's desire for education is ridiculed or rejected.

Elle-Marja's mother speaks to her children only in Sámi and demarcates Sámi culture as home. In contrast, *Elina* is set one generation later in 1952, just before the law forbidding the use of corporal punishment or withholding food was passed. Elina's mother has gone through the same school system and is a fluent Swedish speaker. She speaks both Swedish and Meänkieli/Finnish to her daughters and encourages her children to accept their teachers' demands. Unlike the unnamed teacher in *Sámi Blood*, Miss Holm is unkind and deliberately belittles the children and even the new teacher, Einar (whose last name is never revealed). She clearly states that good education can lift the children out of poverty, and recognizes the importance of good food for nurturing their bodies, all of which indicate a more progressive view of their capacity for growth than Elle-Marja's teacher. However, the simplistic contrast of Miss Holm with Einar erroneously suggests that enforcing Swedish was a choice made by individual teachers, rather than an official, state-endorsed language policy.

In these two films, the prohibition against using local and indigenous languages at school signals the presence of racism in the education system. As Philip Nel notes, "race is present especially when it seems to be absent" (Nel 2017, p. 4). His discussion highlights practices of removing characters of color from places where they should be present. The absence of the ethnic language is a symptom of the ever-present racism. The first step towards acknowledging this form of racism is to expose the history explaining why knowledge of these languages may no longer exist. The right to inform the majority population about the abuses of the past is part of the compulsory curriculum in Swedish schools and considered vital to citizenship education. How this should be taught is not specified, but the films we cite are shared in social media as suggested material. Consequently, we read these films as a broader expression of this policy shift.

The impact of forcing a colonial language on a nation through education and national infrastructure has been extensively examined in relation to the use of English in residential schools for First Nations' children in Canada and the Stolen Generation of Aboriginal children in Australia. These theorizations tend to build on postcolonial theory developed in countries that are no longer under colonial rule, such as Kenya, Nigeria, and India. Some caution must be taken when applying

these theories directly to the Swedish Arctic. Firstly, many Sámi (like many First Nations' citizens and Aboriginals) do not think that the period of colonization has ended: the Finnish Sámi poet, Niillas Holmberg, for instance, considers his home to be in "Saameland, occupied by Finland" (Holmberg 2015). Secondly, when critics such as Ngũgĩ wa Thiong'o, Homi Bhabha, and Gayatri Spivak foreground connections between race and language, they assume that race is visible through skin color. In his seminal essay, "Of Mimicry and Man", Bhabha describes the 'mottled' nature of the mimic to indicate that, no matter how well the mimic apes the colonizer s/he remains "*Almost the same but not white*" (Bhabha 1984, p. 130; italics original). This is not the case in the films we discuss: both the national minorities and national majority are White. In a scene in *Sámi Blood*, one of the SIFR people is amazed by the fact that some of the Sámi pupils are very blond, and in Uppsala Elle-Marja 'passes' as a Swede until she is 'outed' by Niklas. Ethnic affiliation can be signaled through clothing, but it is mainly signaled through language. Without the language, the individual can be so well assimilated that even their children may not know their ethnic background. Elina has assumed that her father was from the Torne Valley, and only learns that he was a Finn at the end of the film. The metonymy of presence is absent; without language, the colonized are invisible even to themselves. The framing narrative of *Sámi Blood* highlights this invisibility: it is not clear whether Elle-Marja's son knew that his mother (and thus he himself) was Sámi whilst he was growing up. As an adult, he seems eager to reconnect with his mother's relatives. He knows a few words in Sámi and uses them when he tries to convince his mother to participate in the reindeer earmarking, as an emotional symbolic linguistic action.

The interactions between Elle-Marja and her son, as well as those between Elle-Marja and her family of origin, highlight the role of shame in internalizing national policy. Elle-Marja endeavors to hide her ethnicity, especially when she interacts with the majority population at a dance and later when she goes to Uppsala. As an adult in the frame story, she denies that she has anything to do with the Sámi. Elle-Marja's shame is destructive, not least because of the divisions it causes between her and her immediate relatives. However, Ahmed (2014) has identified shame as a critical emotion in the work of apology. For Ahmed, recognizing shame comes with limits and conditions that guide the interpretation of narratives. Viewers of *Sámi Blood* need to recognize that Elle-Marja's feelings of shame stem from the colonization process: she has committed no crime for which she should be ashamed. On the contrary, she should have been able to feel pride in her knowledge of two languages and cultures. In recognizing Elle-Marja's feelings of shame as misplaced, viewers are positioned to assume the shame themselves. As Ahmed explains, shame "becomes not only a mode of recognition of injustices committed against others, but also a form of nation building" (p. 102). By watching films depicting shameful histories, Swedes are encouraged to acknowledge a collective guilt, thereby redirecting Elle-Marja's feelings of shame onto the more appropriate target: those who have benefitted from the colonization process. This transference is morally right, as Ahmed explains, but it is also complicatedly implicated in the work of nation building. A nation that aspires to uphold human rights needs to be troubled by their own state-endorsed injustices in the past.

Restoring relations between majority and minority populations or reconciling the Swedish nation with its shameful past creates a sense of national unity, since "feeling bad" about national transgressions can enable feelings of "coming to terms with" the guilt (Ahmed 2014, p. 102). Witnessing past injustice, and the failure of the nation to live up to its ideals, "becomes the ground for a narrative of national recovery" (Ahmed 2014, p. 109). By funding projects, such as the films we discuss, which encourage viewers to feel hurt and shame, the nation enacts an apology via substitution: "the expression of shame can be a substitute for an apology, while an apology can be a substitute for shame" (Ahmed 2014, p. 120). Nevertheless, this process is complicated by its connection to nationalism: acknowledging the collective failure to live up to its ideals effectively proposes that because "*we mean well*, . . . [we] can work to reproduce the nation as an ideal" (Ahmed 2014, p. 109; italics original). To be appropriate, a pedagogy of apology needs to negotiate a balance between enabling genuine apology to take place without promoting the kinds of nationalism that supported the original abuse.

In both films, the role of schools in promoting a national ideal is laid bare. The role of inspiring feelings of national unity through collective guilt (and, perhaps, collective victim status in the case of national minority viewers) is less clear. In *Elina*, feelings of shame are altogether rare: Elina's refusal to feel guilt or shame are presented as admirable traits. Initially, Miss Holm is also depicted as though she thinks herself detached from shame; she considers it to be her duty as a teacher to punish pupils for speaking their own language or for using their left hand when writing. In Ahmed's words, "detachment of shame from individual bodies does a certain kind of work with the narrative" (p. 102). The viewer will sympathize with Elina, and feel shame on the part of Miss Holm, the representative of the nation to which the viewer is expected to feel allegiance. As the story unfolds and Miss Holm finally apologizes and begs forgiveness, Elina nods her acceptance of the apology. This locates guilt and shame in one individual teacher rather than within a system; it does not promote collective shame. It also suggests that standing up to the system was a matter of individual character, which implicitly leaves responsibility for resisting colonization on the small shoulders of school children. Furthermore, the viewer might feel relieved of the shame, and "*the nation can 'live up to' the ideals*" (p. 109; italics original), in which case there can be no real reconciliation.

As mentioned, *Elina* is a children's film. The concretization of major issues as individual acts is likely to be easier for a child to comprehend, but from a pedagogical standpoint it is deeply problematic. *Sámi Blood*—addressed to an adult audience—proffers opportunities for more sophisticated interpretations. Whilst the teenage Elle-Marja is easy to like and admire, the adult Elle-Marja in the frame story comes across unlikeable. Her insistence that she has nothing to do with the Sámi comes across as rude and alienating: she calls them thieves and complains about their shrill music. She rejects gestures of friendship and is impolite to everyone. However, we also see her ashamedly pulling at her hair to hide her damaged ear, cut by Swedish boys when she was a teenager, comprehending Sámi and other details. Viewers are encouraged to recognize that the adult Elle-Marja's cantankerous behavior is a result of her adolescent experiences. At the end of both films, both protagonists are shown using their ethnic languages again. However, where Elina is depicted as being deeply connected to the land itself, Elle-Marja's relationships with the Sámi and Sápmi remain ambiguous.

## 3. The Role of Language in Resisting Colonization

Writing from the Kenyan context, Ngũgĩ articulates the need to write in local and indigenous languages as a means of "Decolonising the Mind" (Wa Thiong'o 1986). Ngũgĩ's point is to highlight the internalization of the colonial mindset implicit in the language and the value of resistance through using local and indigenous languages. The language policy in Sweden in the first half of the 20th century was intent on undermining local and indigenous languages, and consequently the cultures of historically established minorities. As *Elina* and *Sámi Blood* show, teachers enforced the use of Swedish at the expense of the children's cultural heritage. Dei and Simmons, writing about the current situation in Ghana, highlight the disciplinary nature of these practices.

> Often, citizenry and community, as being produced by way of schooling and education through various rituals of belonging, come to be engendered through ethnocentric curricula and pedagogies. Within the culture of schooling, the sense of community and the self come to be disciplined through these regulatory procedures of citizenship. The classroom then—in and of itself—comes to exert pedagogic forms of discipline onto students, thus addressing the needs concerning responsibility, the emotional, and the social, as framed within the context of the different local communities in a particular way that encumbers embodied ways of knowing. (Dei and Simmons 2016, p. 4)

The classrooms in both the films concretize Dei and Simmons' points. Both Elle-Marja and Elina are good students who are functionally bilingual: they represent the best possible outcome of the school system. By exposing the price paid for their success, the films spit in the face of

those who would claim success for the Swedish-only policy. Although Elle-Marja and Elina are both fluent in Swedish, they prefer to use their home languages. The fact that they are both girls is significant. As Spivak explains in her discussion of the colonized subject (the 'subaltern'), "If, in the context of colonial production, the subaltern has no history and cannot speak, the subaltern as female is even more deeply in shadow" (Spivak 1988, p. 287). Rather than celebrating the girls' successes, the classroom exerts pressure that divides communities and even families, without offering opportunities for true emancipation.

This lack of emancipation is more evident in *Sámi Blood*, set in the 1930s when Elle-Marja must choose to be either Swedish or Sámi. She rejects her ethnic identity, symbolized through the sale of her inheritance—reindeer and a silver belt—to pay for her education. Further rejections of her cultural heritage are expressed through burning her Sámi clothes, changing her name, and insisting that she does not speak Sámi. Elle-Marja is a Swedish citizen, but as a Sámi, she is denied basic rights. Her rejection of her cultural heritage is a concretization of the colonized mind. In the contemporary setting, she can be both Sámi and Swedish. This is symbolized in the burial held in Sámi, where Christina/Elle-Marja wears a non-Sámi dress but is still part of the cultural community, shown by the fact that she is addressed in Sámi by her sister's husband. However, she is not released from her feelings of shame until she apologizes to Njenna's corpse in Sámi some hours after the funeral. After this apology, Elle-Marja leaves the church and clambers up a muddy hill to join the Sámi as they gather to mark the reindeer calves' ears. Splattered with the mud of Sápmi, she gazes at the scene where the Sámi are depicted using traditional tools (a lasso and knife) alongside helicopters, motorbikes, and all-terrain vehicles. Some wear traditional clothing, some in combination with practical outdoor sportswear. The camera follows Elle-Marja's gaze but leaves it for the viewer to interpret the scene. The Sámi have modernized without relinquishing all their traditions. Today one can be both Swedish and Sámi but, in the 1930s, Elle-Marja could not. Whether she can regain her heritage is unclear, but the final scene suggests a willingness on both her part and from the community.

In the children's film, *Elina*, the importance of language in resisting colonization is simplified to the point of didacticism. Elina speaks Swedish better than the other pupils (partly because she is older). On the first day of school, another pupil, Anton, asks Elina for a Swedish word so that he can complete his assignment. Miss Holm publicly humiliates Anton for using Finnish and decrees that he may not eat lunch as punishment. Elina attempts to defend him, but is silenced. She then offers Anton her own lunch, and when she is rebuked, she refuses to eat it herself. This event is repeated over the following days. Since Elina is defying the teacher's instructions, Miss Holm declares that "from here on she is air to me", which provides the film with its title: Miss Holm behaves as though Elina wasn't there. She closes the classroom door in Elina's face, ignores her knocks, and behaves as though Elina were invisible. (The Finnish title of the film is *Invisible Elina*). The title emphasizes the idea that Elina is not seen, but the storyline focuses on the idea that she is not heard, despite her excellent language skills. Matters come to a head when Elina runs away from the school to find solace in the fen and begins to sink. She is rescued by Irma, her mother, and Einar. The following day at school, Elina refuses lunch again. She is first joined by Irma, then Anton, all the other children, and Einar, who greets the children in Finnish. Finally, Miss Holm apologizes for being unfair to Elina. She invites them to return to school and eat their lunches. The children oblige.

As the above summary shows, *Elina* ahistorically suggests that resisting language colonization was primarily a matter of individual free will. The street-level bureaucrats—Miss Holm and Einar—are portrayed as individuals, not as representatives of a more powerful machinery. Indeed, the inclusion of a teacher who shows respect for the children's language and local knowledge, ahistorically suggests that Miss Holm's implementation of the Swedish-only policy of the era is a reflection of her personal unpleasantness. Einar's role as a White savior is both problematic and ahistorical. The nameless teacher in the more historically accurate *Sámi Blood* is closely aligned with the machinery of colonization, symbolized through the 'scientists' who come to photograph the pupils. Her lack of a name further encourages us to perceive her as part of a system rather than as an individual. When she tells Elle-Marja

that her brain is not capable of the studying needed to become a teacher, she believes this is a neutral, scientific fact. Resisting colonization requires standing up to far more than an individual teacher's mind-set: it requires a community to resist an ideology.

## 4. Resisting Colonization

As already noted, *Elina* ends happily when the whole school, including Einar, comes together to defeat Miss Holm. It is so historically implausible that we focus our attention in this section exclusively on discussing the more realistic problems Elle-Marja faces. Not only must she overcome systematic abuse within the school system, but she must also cope with abuse from the local community. The village youths mock the Sámi pupils as they are marched from their dormitory to the schoolroom and back. On one occasion, Elle-Marja responds by asking whether they are good enough to receive special visitors from Uppsala. Elle-Marja is proud that she has been chosen to recite a poem for them. Her pride turns to humiliation as the visitors perform phrenological examinations and photograph her naked body. The local boys watch this indignity through the window, and jeer when the Sámi children are marched back to the dormitory. Elle-Marja grabs the ring-leader and demands an apology, but she is overpowered and earmarked like a reindeer. Unlike the photography, this physical violation is not sanctioned by the school, but the boys go unpunished.

The significance of the attack requires some historical background. The first reindeer husbandry law was passed by the Swedish parliament in 1886. This Act decreed that only those Sámi who earned their living through reindeer husbandry were "real" Sámi. The Act Concerning the Rights of Swedish Lapps to Reindeer Herding (Lag 1928:66) further limited reindeer husbandry rights. Even today, reindeer husbandry requires membership in a Sámi community. The ear-markings on the reindeer are inherited. Gathering to mark the new calves' ears is probably the most important festival in the Sámi year. Elle-Marja fulfills the criteria in the 1930s for being considered a Sámi, but as a pupil at the residential school for Sámi children, she is not considered equal to Swedish pupils. At the end of the film, viewers learn that her sister continued to use Elle-Marja's mark on new generations of calves. She is thus unquestionably still a "real" Sámi. The young man who marks her ear, however, does not have this right. He could be of Sámi origin, but a descendant of a settled Sámi who was not considered "real" Sámi. This would explain how he knew the marks. Equally, he might simply be cutting randomly. The audience must interpret the words of the protagonist—"You are a Lapp as much as I am", and the boy's reaction. Either way, the attack makes it clear why Elle-Marja feels she must leave her northern home. Pulling her hair to hide the scars on her ears becomes a leitmotif throughout the film for reminding viewers of the prejudices beyond the school system. Indeed, the only occasion in which she is depicted as belonging to the Sámi community is when she sings a joik[2] to her sister on their way to school.

Prior to the attack, Elle-Marja has attended a dance. A conscript soldier from Uppsala, Niklas Wikander, dances with her, and tells her about his home. After the attack, she takes a train to Uppsala, where she finds Niklas in his parents' home and initiates sex thereby securing a place to spend the night. Afterwards, we see her gazing at sperm fluid—the source of their genetic difference—between her fingers. She is clearly fascinated; her response is playful and empowered. When Niklas's parents object to Elle-Marja staying in their home, Elle-Marja seeks refuge in parks, where she lies down on the lawn, pressing her cheek against the grassy surface as though seeking a connection to the ground. The use of long shots and shadows signal her vulnerability: she does not belong in this human made environment. In contrast, when Elina adopts the same position in the fen, she appears to connect with her father's spirit. The camera draws back allowing us to view Elina blending into her environment,

---

2　'Joik' is a traditional form of Sámi song that creates "a feeling of unity within the group . . . it reinforced his or her sense of identity as a member of a family and community" (Gaski 1999, p. 34). They are typically sung to evoke a place, a person or an animal.

her clothing connecting her to the colors of the surrounding plants. Despite her initial experiences of alienation, Elle-Marja decides to study in Uppsala.

*Sámi Blood* critiques Swedish colonialism by depicting assaults on the Sámi people by the Swedish state and individuals empowered by the state in a geographical area where Swedes were traditionally a minority. The use of the Sámi language adds credibility as it contributes to decolonization of the mind, if not for viewers from the national majority, at least to Sámi viewers. The film also problematizes the unwillingness of the Sámi to accept Elle-Marja's desire for a different life and the internalization of racial prejudices. Thus, the film is not only a postcolonial critique of the politics of the Swedish state, or a document of Sweden's shameful past, it also advocates the right of an individual to be Sámi in her own way. Identifying with two cultures can be seen as an expression of post-colonial hybridity. Unlike the opening towards apology in *Elina*, *Sámi Blood* does not proffer any "ground for a narrative of national recovery" (Ahmed 2014, p. 109). No helping hand is outstretched, and no apology offered or accepted.

## 5. Belonging to the Arctic

Both films celebrate the protagonists' connection to the land: they speak the language of the place they inhabit, and are able to interpret the natural elements more effectively than the Swedish characters. *Elina* is set in a region dominated by treeless fells, fens, and the great Torne River. The opening scene depicts the fen using scans to indicate that the viewer is viewing the landscape through Elina's eyes, as she calls to her deceased father in Finnish saying "I know you are here". The audience then see Elina skipping across the fens repeating her father's phrase: "remember to move, do not stop". The point being that one needs to keep moving across fen land or one will sink. This refrain functions as a leitmotif both for recalling Elina's sense that her father's spirit is alive in the fens and for highlighting the ever-present danger of drowning. The scene ends with Elina returning home where her mother scolds her in Swedish: "The fens are no place to play!" Thus, Elina is associated with the landscape where she feels connected to her father. *Sámi Blood* also begins with a sequence of shots depicting Elle-Marja and Njenna bidding farewell to their mother beside a traditional *kåta* (a Sámi tent). The girls' journey to school combines wide shots situating the pair in an open fell landscape. Their journey ends when they row across a lake, whilst Elle-Marja joiks for Njenna. Elle-Marja warns Njenna not to joik in school. No threat is evident in *Sámi Blood*'s opening scenes; it emerges when the girls are herded past the village boys on the road to school and, as noted above, this is where Elle-Marja will be treated as an animal and earmarked. Elina's village appears to be a safer environment; the villagers speak Meänkieli among themselves. However, the road through Elina's village, like the road in *Sámi Blood*, is a site of danger.

In "Forms of Time and of the Chronotope in the Novel", Mikhail Bakhtin drew attention to the fusing of time and place in the road, in the form of chance encounters (Bakhtin 1982, p. 243). The road is a shared space where time determines who will meet whom making it a place of opportunity, but also danger. Both films exploit this trope. In *Sámi Blood*, Elle-Marja is attacked on the road, but her train journey brings new opportunities. The road is also a place where she can shift identity. On the way to the dance, she steps off the road to remove her Sámi clothing and put on her stolen dress. In doing so, she actively chooses how to represent herself. In *Elina*, the road allows for several chance encounters, the most significant involving the new teacher, Einar. He offers Elina and her mother a lift, but the car fails. He asks Elina to start the car whilst he fixes the engine, but the car begins to roll away with Elina at the wheel. Einar manages to stop the car before it hits Miss Holm, but the road is a place of danger. The film ends suggesting that the road is also a place of opportunity: Einar offers another lift home in his newly repaired car. Einar is clearly attracted to Elina's mother, and although she is not ready to return his feelings, there are suggestions that she might do so in the future.

Since Elina moves between her home, the fens, the road, and the school throughout the film, the interplay between the types of knowledge needed in the different environments is easy to see. Elina's engagement with the muddy fen waters, with its synechdochic ties to her father, symbolize her

relationship with the landscape she loves most. Her ability to read the watery landscape is valued by her community. When a calf gets stuck in the fen, the farmer asks Elina for advice. The scene reveals that Elina's knowledge of the fens is extraordinary. Later, Einar also seeks her out for her knowledge of the land as he asks her advice on picking mushrooms. He is out of place in the landscape, but is willing to learn.

The fen mud also connects Elina's family. The physicality of the lived landscape creates unity. However, when Elina runs away from school to hide in the fens, she forgets her father's warning to keep moving and so comes home with filthy, ruined shoes. The mud still represents her father, here in his stubbornness and his insistence on returning to the fen, and also shows how school is reducing Elina's contact with the land as she moves awkwardly. The conversation shifts between Swedish and Finnish. The mother greets her with questions about school in Swedish (the language of school), but shifts to Finnish (the language of the fens) when she sees the muddy shoes and returns to Swedish when she explains their need to ask Miss Holm for assistance. The fens become a danger for Elina when the world of school drives her into them. Distraught at being treated "like air", Elina runs off to the fens and forgets the message of her father's song: she stops moving and starts to sink. Irma fetches her mother and Einar, who arrives with the school noticeboard to spread the weight and enable them to pull her out. Once again, mother and daughter are covered in mud, reunited in an embrace.

The final scene depicts Elina in the graveyard with her mother and sisters. She learns that her father was not local, and that her mother taught him about the fens. Thereafter, he always took Elina into the fen, whilst his wife stayed at home. Elina finally says farewell to her father. In *Sámi Blood*, there is a parallel, in which the ageing Elle-Marja comes to her sister's funeral. After refusing to take part in any Sámi activity, or even acknowledge that she understands the Sámi language, she visits the church, lifts off the lid of the coffin, and asks her dead sister to forgive her. Then she goes up into the mountain, walks into the Sámi village, which has modernized during her absence. While Elina has remained in her Arctic belongings, Elle-Marja returns to her Arctic homeland, although it is not clear whether she will choose to rebuild her connections. Whereas language and cultural heritage are under threat from the Swedish state, the physical territory cannot be taken away from the protagonists. In the case of Elle-Marja, she rejects her Sámi heritage and thus becomes alienated. This is evident in Elle-Marja's muddy climb up the hill to the see the village. Her inappropriate shoes cause her to slip and slide as she climbs: she is no longer at home. The mud that connected Elina and her family is used in *Sámi Blood* to show that Elle-Marja is disconnected.

## 6. Citizenship Education: Situating the Films in the Time of Their Production

*Elina* and *Sámi Blood* are two contemporary films with common denominators. The two protagonists carry the double burden of being members of national minorities, and females (Spivak 1988). They are subjected to Swedish supremacy and are expected to retain their marginalized position, despite their excellent language skills. Elina is treated as though she did not exist, and Elle-Marja is forced to deny her Sámi cultural heritage, her name, and her language in order to be educated. While *Elina*, a film intended for a younger audience, suggests that all that is needed for reconciliation is an apology, *Sámi Blood* insists that there is more work to be done. The difference is not solely due to the intended audiences. *Elina* is based on a book by the non-minority Swedish author, Kerstin Johansson i Backe, published in 1978. Klaus Härö, the director, and the actors were not Tornedalingar, thus *Elina* reflects the view of outsiders. Amanda Kernell, the writer and director of *Sámi Blood*, has Sámi roots as do the key actors, making the perspective closer to that of insiders. These factors contribute to the differences in the way national shame and apology are linked in the two films.

In both cases, the films depict genuine historical events: the prohibition to speak any other language than Swedish in schools and the poor quality of residential schools. These films from the 21st century must also be seen in relation to the National Minorities Acts. *Elina* was released seven years before the first National Minorities Act, and *Sámi Blood* was released six years after it. This Act enshrined the right to inform the national majority about discrimination, both institutional

and individual, past and present in law. Consequently, *all* pupils in Swedish schools must be taught 'about' the national minorities, but precisely what pupils are to be taught is determined by individual teachers. Despite the lack of specifics, these inclusions can be considered part of a broader policy of education related to democratic citizenship. Sweden's commitment to democratic values and, more specifically, democratic citizenry is implemented throughout the education system with audits, carried out by the Swedish Schools Inspectorate. In addition, museums and publicly funded artistic endeavors, including film production, must support the enactment of the National Minorities Act. *Elina* and *Sámi Blood* both received public funding. Consequently, it is reasonable to claim that these films are part of an overt, national policy aiming at citizen education and promoting democratic values.

For viewers from the majority population, these films may be seen as narratives of injustice carried out in the past against national minorities. If so, the films may contribute to a national reconciliation, through feelings of shame (cf. Ahmed 2014). By watching the films, modern day viewers will face the narrative of oppression and injustice, and will have to accept that the Swedish state was not well-intentioned. However, as Ahmed (2014) points out, the admission of guilt and the feelings of shame, may also have a reconciliatory function. Modern day viewers may tell themselves that "it is no longer like that". As noted above, viewers from the national minorities might, however, consider that the struggle continues. In the films the protagonists hold different positions, but they are positioned *beside* representatives of their cultural heritage and of the Swedish state. Sedgwick (2003) explains the notion of *beside* as a dualistic phenomenon that

> comprises a wide range or desiring, identifying, representing, repelling, paralleling, differentiating, rivaling, learning, twisting, mimicking, withdrawing, attracting, aggressing, warping, and other relations. (Sedgwick 2003, p. 8)

In the case of *Sámi Blood*, being positioned *beside* comprises several of the relations mentioned by Sedgwick. As a Sámi, Elle-Marja represents her linguistic and cultural heritage, down to the obligation to wear Sámi clothing. At the same time, the authorities reject her heritage by forbidding her to speak Sámi. Their school mimics mainstream Swedish schools, but does not provide entry into further education. In order to attend the dance or the school in Uppsala, Elle-Marja must not be identified as a Sámi. Every interaction with the Swedish society reinforces this message. In essence, Elle-Marja holds a position *beside*: Swedish on one side, Sámi on the other.

Both films clarify how the language policy destroyed communities and romantically intimate that resistance is possible through an intimate knowledge of the land. Knowing the land allows the widows, Elina's and Elle-Marja's mothers, to feed their daughters. Knowing and acknowledging the land also makes resistance possible. Truth and reconciliation may begin with acknowledging abuses of the past, but revaluing people who have been systematically abused is also necessary. As of 2019, all curricula from pre-school, all the way through high school, require that pupils learn about indigenous people and national minorities, their history, culture, and language. However, this is only a first step. Reconciliation demands that shame—both the shame of the oppressor for acts of suppression and the undeserved feelings of shame still experienced by those who were oppressed—be acknowledged. By achieving this recognition, *Elina* and *Sámi Blood* contribute to a citizenship education.

**Author Contributions:** Conceptualization, L.K., A.P. and L.M.; Investigation, L.K., A.P. and L.M.; Writing–original draft, L.K., A.P. and L.M.

**Funding:** This research received no external funding

**Conflicts of Interest:** The authors declare no conflict of interest.

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
