# Peer review of "Protest and Apology in the Arctic: Enacting Citizenship in Two Recent Swedish Films"

_humanities, doi:10.3390/h8010049_

Round 1

Reviewer 1 Report

The bulk of the introduction is taken up with presenting historical context information regarding the Swedish national minorities. The most specific problem statement is this: ‘we seek to examine how these films contribute to the work of apology’ (for past institutional mistreatment of national minorities), lines 84-85.  There is no outline of the methodology forming the basis of the examination.

The article interestingly discusses the two films in relation to the Swedish historical past, but the lack of specificity regarding the analytical framework carries through the article, which retains the historical angle more strongly than it develops any film-aesthetic angle. How are the films analyzed? ‘The use of long shots and shadows signal her vulnerability’ (lines 315-316) is the only attempt to combine film aesthetics with the interpretive comments regarding the films that are liberally made in the rest of the text.  The article would benefit from a clarification of its methodological and analytical framework. It does as it promises, however, and ‘examines’ the films in the context of national apology.    

Author Response

Review Report Form

Open Review

English language and style

( ) Extensive editing of English language and style required 
( ) Moderate English changes required 
(x) English language and style are fine/minor spell check required 
( ) I don't feel qualified to judge about the English language and style 

Yes

Can be improved

Must be improved

Not applicable

Does the introduction   provide sufficient background and include all relevant references?

( )

(x)

( )

( )

Is the research design   appropriate?

( )

(x)

( )

( )

Are the methods adequately   described?

( )

( )

(x)

( )

Are the results clearly   presented?

( )

(x)

( )

( )

Are the conclusions   supported by the results?

(x)

( )

( )

( )

Comments and Suggestions for Authors

The bulk of the introduction is taken up with presenting historical context information regarding the Swedish national minorities. The most specific problem statement is this: ‘we seek to examine how these films contribute to the work of apology’ (for past institutional mistreatment of national minorities), lines 84-85.  There is no outline of the methodology forming the basis of the examination.

Response: We have put the sentences together more closely to make it easier to identify.

Our aim is to examine how these films contribute to the work of apology as they portray female characters who demonstrate agency through their knowledge of the Arctic landscape and their resistance to school policy. We begin by outlining how the films portray the Swedish state’s attempt to undermine minority cultures within the Arctic region through the school system. We then resituate the discussion considering how the films function in the historical context of their production: one six years before the passing of National Minorities Act (Elina), the other seven years after it was passed (Sámi Blood).

The article interestingly discusses the two films in relation to the Swedish historical past, but the lack of specificity regarding the analytical framework carries through the article, which retains the historical angle more strongly than it develops any film-aesthetic angle. How are the films analyzed? ‘The use of long shots and shadows signal her vulnerability’ (lines 315-316) is the only attempt to combine film aesthetics with the interpretive comments regarding the films that are liberally made in the rest of the text.  The article would benefit from a clarification of its methodological and analytical framework. It does as it promises, however, and ‘examines’ the films in the context of national apology.  

Response: as the reviewer observes, our main analytical framework is indeed historical and not film-aesthetic. Most Swedes are not aware of this history. It has recently been added to the national curriculum and we teach it to student teachers, many of whom do not even know they are members of a national minority prior to the course! We cannot expect that the international readership of Humanities would know. The framework resonates to two of the stated analytical frameworks in the special call:

n  children’s literature as national and transnational heritage in institutional contexts

n  children’s narratives as materials for citizenship education     

Submission Date

07 January 2019

Date of this review

01 Feb 2019 12:19:52

Reviewer 2 Report

A well written, timely, significant, comparative analysis of two important, recent Swedish films representing the education of indigenous and minority groups in relation to enforced separation of children and youth from their home cultures and families through language teaching and alienation from nature.  Effective use of Ahmed in particular.  The article is very good--as far as it goes.  However, it can be made even better by unpacking some concepts, providing more explicit evidence, and interrogating some key assumptions.

Effective structure of article in showing striking similarities and significant differences between these two films.  However, two key issues appear late in the article almost as throwaways because their significance is not unpacked.  Both abstract and introduction should prepare readers for: 1) fact that one film is a "children's film" and one an "adult film," that one features a young child while the other features an adolescent (and her older self); 2) that one film was created by creators outside the cultural group represented while the other was by "insiders."  Both of these are worth flagging from the outset and unpacking throughout.  (Be careful, though, about making "common sense" assumptions about "children" and "adults" and "outsiders" and "insiders."  For example, I'd question your assumptions at 214-17 about "undoubted" differenceas between children and adults.)

Identify whether there are any significant distinctions to be made between "indigenous" (Sámi) and other "national minorities" (e.g. the Tornedalingar).

"Feisty female" (repeated) seems cliched and may ignore significant differences between your two protagonists.  A more nuanced description of the protagonists would be helpful than this "shorthand" for "female agency": in what ways are these characters strong, assertive, independent, courageous, intelligent, willful, stubborn, persistent, etc.?

Be even more explicit about when, where, and to whom characters speak specific languages.  For example, Elina speaks Finnish, Meänkieli, and Swedish: how and when and why does she deploy each in the film? 

Beware of exaggerated or unsubstantiated claims: for example, 64-65 "it is no exaggeration to state that colonial practices in the Arctic region constituted a form of ethnic cleansing."

The roles of the church and Christianity in state-sanctioned education is not mentioned though these seem to figure in both the films and historically.  (The children sing a hymn linking Christian fathers to Sweden: does God speak in Swedish only?)

Some of your arguments risk an oversimplifying essentialism: these people belong to the land and the land belongs to them and if only they could go back before colonial contact stole their languages everything would be find.  For example, one could argue that Elle-Marja's "agency" is demonstrated (rather than stolen from her) by the various choices she makes in her life, including those to reject narrow conceptions of her Sámi heritage.  Her story and the film are complex as are her choices, and to imply that her clambering up the hill at the end resolves everything by undoing the rest of her life and restoring it to an Edenic simplicity does neither justice.  (The Swedish tourists complaining about the noise of the "motorbikes" disturbing this "nature reserve" and lamenting, "I thought Sámi were supposed to be close to nature" represents such simplistic thinking.  Note that the Sámi at the end of the film are using helicopters and all-terrain vehicles to go mark the ears of the reindeer.  Elle-Marja does ask her sister to forgive her at the end, but that doesn't necessarily undo her years of changing her names, dress, and attitudes.  Unless, of course, we assume that she's simply awakening from years of "false consciousness": but isn't that as condescending as her teacher telling her that she can't become a teacher because her family needs her, she needs to use her skills on the land, and she's not smart enough to become a teacher?)  What is perhaps underemphasized in your argument is a sense of the complexities of postcolonial hybridity.  In other words, does Elle-Marja lose her status as "real Sámi" when she is in Uppsala and "southern Sweden"?  Is she only "real Sámi" when she is with the reindeer on the land?

There are some seemingly powerful symbolic moments which could be analyzed and explicated more to good effect.  For example, 266 - what is the symbolic significance of Miss Holm behaving as though Elina is invisible?  For example, 383-84 - "She learns that her father was not local, and that her mother taught him about the fens."  For example, characters in both films holding their ears to the ground.  For example, the poem recited by Elle-Marja's teacher and her by Swedish-speaking Finnish poet Edith Sodergran, "I long for the land this is not."  Significance of names "Christina" and "Niklas"?  In Elina, the teacher repeatedly changing Anton's pen from his left hand to his right hand.

The character of Einar in Elina perhaps deserves some skeptical resistance.  Does he oversimplify the narrative (for children) by being the "good cop" to the female teacher's "bad cop"?  He wants to learn language from his students, goes out in solidarity with them, has eyes for the protagonist's mother, "rescues" Elina when her mother's apron isn't enough, drives off into the sunset with Elina and her mother at the end, giving them a ride from Elina's father's grave.  Is this "white knight" from southern Sweden a benign saviour for the northern women who can't quite make it on their own?

Some specifics:

31 - "their ethnic minorities" rather than "its ethnic minorities"

36 - I believe the English title is Elina: As If I Wasn't There while the French title is L'Invisible Elina.

63-64 - "a gap . . . was significant" vs. "were"

83 - "we do not intend to conflate the Sámi and the Tornedalingar" - Might you unpack this more by making a distinction between "indigenous" and other "national minorities"?

110 ff. - here and later, you could make your "close reading" of this crucial part of the film more explicit (and hence more powerful): the violence of the flash of the photographer's camera; the specific measuring instruments (including the examining of the teeth inside the mouth--are reindeer measured in such ways?: horses are); the voyeurism (let's take a picture of the two of them holding hands); the enforced nudity (put your hands behind your heads); the complicity of the teachers).

153 - a mention of Canadian residential schools would add to this list (there are many overlaps, particularly with residential boarding schools which took children from their parents and communities with the intent of language re-education as well as industrial training)

176 ff. - Some specific quotations from the older Elle-Marja would make this come alive even more (She complains about "those people" lying, stealing, whining, their music is "shrill," "fucking circus animals," etc.)

183-84 - This is crucial: it could be unpacked more.

217 ff. - I'm not convinced that "whilst the teenage Elle-Marja is easy to like and admire, the adult Elle-Marja in the frame story is deeply unlikeable."  Is not her "agency" as both adolescent and adult admirable?  It's worth noting that both the adolescent and the old woman pull a lock of hair over their wounded/marked ear.  Your argument seems to imply that Elle-Marja's whole adult life has been forced upon her and is redeemed only when she reconnects with the land, her language, her heritage.

275 - "free will" vs. "freewill"

313 - "trades sex for a place to spend the night": This seems a misrepresentation of what the film shows, namely, that EM seeks out Niklas at the dance, then goes to his home, then asks if she can stay, and then they have a consensual sexual relationship.  Even if he turns out unwilling or unable to stand up to his parents and friends, she has demonstrated her agency in initiating this relationship, has she not?

333-34 - "Both films . . . suggest that agency within the Arctic landscape should determine who belongs there."  The implications of this claim would seem to open up "might makes right" and other unfortunate possibilities.  If only a minority of Sámi today live a traditional lifestyle on the land, does that make them the only "real Sámi"?

442-45 - Is Elle-Marja's agency and resistance possible only through "knowing and acknowledging the land"?  Does she not demonstrate resistance both from her Swedish teachers and education and from her Sámi family and community?  Is she less Sámi in a library reading, reciting a Swedish poem, lying in an Uppsala park, changing her names and her clothing, staying in a hotel instead of a tent?  Is she "real Sámi" only when she is near reindeer?  How can one do justice to the complexity of her agency while still recognizing and exposing those things that were forced on her (including the enforced marking of her ear--which, as you note, may problematically be done by a fellow Sámi).  Does she not resist both her mother's "Speak Sámi" and her teacher's "Speak Swedish"?  Does she not live her life--as much as is possible for anyone--in a both-and rather than an either-or?

Finally, thanks for the article and for sharing two wonderful films with others!

Author Response

Review Report Form

Open Review

English language and style

( ) Extensive editing of English language and style required 
( ) Moderate English changes required 
(x) English language and style are fine/minor spell check required 
( ) I don't feel qualified to judge about the English language and style 

Yes

Can be improved

Must be improved

Not applicable

Does the introduction   provide sufficient background and include all relevant references?

( )

(x)

( )

( )

Is the research design   appropriate?

(x)

( )

( )

( )

Are the methods adequately   described?

(x)

( )

( )

( )

Are the results clearly   presented?

(x)

( )

( )

( )

Are the conclusions   supported by the results?

( )

(x)

( )

( )

Comments and Suggestions for Authors

A well written, timely, significant, comparative analysis of two important, recent Swedish films representing the education of indigenous and minority groups in relation to enforced separation of children and youth from their home cultures and families through language teaching and alienation from nature.  Effective use of Ahmed in particular.  The article is very good--as far as it goes.  However, it can be made even better by unpacking some concepts, providing more explicit evidence, and interrogating some key assumptions.

Response: Thank you.

Effective structure of article in showing striking similarities and significant differences between these two films.  However, two key issues appear late in the article almost as throwaways because their significance is not unpacked.  Both abstract and introduction should prepare readers for: 1) fact that one film is a "children's film" and one an "adult film," that one features a young child while the other features an adolescent (and her older self); 2) that one film was created by creators outside the cultural group represented while the other was by "insiders."  Both of these are worth flagging from the outset and unpacking throughout.  (Be careful, though, about making "common sense" assumptions about "children" and "adults" and "outsiders" and "insiders."  For example, I'd question your assumptions at 214-17 about "undoubted" differenceas between children and adults.)

Response: We have added this information when we introduce the films and make reference to …. it explain where.

Identify whether there are any significant distinctions to be made between "indigenous" (Sámi) and other "national minorities" (e.g. the Tornedalingar).

Response: In relation to these two films, the most significant point is that Elle-Marja’s mother is a reindeer herder and thus lives a semi-nomadic life (we see her beside her kota). Elle-Marja must attend a residential school. Elina’s parents have a house and so Elina can return to her mother each evening, which makes her ties to family tighter. We have added this information.

"Feisty female" (repeated) seems cliched and may ignore significant differences between your two protagonists.  A more nuanced description of the protagonists would be helpful than this "shorthand" for "female agency": in what ways are these characters strong, assertive, independent, courageous, intelligent, willful, stubborn, persistent, etc.?

Response: We agree. We continue to use the shorthand for the abstract, but thereafter have attempted to be more nuanced, especially in our interpretations of Elle-Marja and her more ambiguous forms of resistance.

Be even more explicit about when, where, and to whom characters speak specific languages.  For example, Elina speaks Finnish, Meänkieli, and Swedish: how and when and why does she deploy each in the film?

Response: We have tried to add a few examples, to the already existing ones. It is not always possible to distinguish between Meänkieli and Finnish. The film basically uses standard Finnish with just a few accented points to show that they are in the North.

Beware of exaggerated or unsubstantiated claims: for example, 64-65 "it is no exaggeration to state that colonial practices in the Arctic region constituted a form of ethnic cleansing."

Response: We don’t consider that an exaggeration and included detailed evidence about reduced life expectancy supporting this claim in the first draft. People dying earlier than they did pre-colonization did reduce the number of people who were alive. It was not random: the life-expectancy of those from Sámi backgrounds was longer prior to 1850 when colonial practices began than it was later in the nineteenth century. Elsewhere in Sweden, life expectancy increased during this period. They were not shot or gassed, but they died due to colonial practices. This is a form of ethnic cleansing. However, the films do not deal with these practices and so we have removed the paragraph.

The roles of the church and Christianity in state-sanctioned education is not mentioned though these seem to figure in both the films and historically.  (The children sing a hymn linking Christian fathers to Sweden: does God speak in Swedish only?)

Response: We did include more of this in our earlier drafts but removed it in order to keep the word count down. Elina includes a scene where her mother meets a village woman and they use a Laestadian-specific greeting (in Finnish). The Bible may be taught in Swedish (Elina learns about the Tower of Babel in school), but God is worshipped in Finnish. At Njenna´s burial Sámi is spoken. A properly nuanced discussion of religion would require several hundred words.

Some of your arguments risk an oversimplifying essentialism: these people belong to the land and the land belongs to them and if only they could go back before colonial contact stole their languages everything would be find.  For example, one could argue that Elle-Marja's "agency" is demonstrated (rather than stolen from her) by the various choices she makes in her life, including those to reject narrow conceptions of her Sámi heritage.  Her story and the film are complex as are her choices, and to imply that her clambering up the hill at the end resolves everything by undoing the rest of her life and restoring it to an Edenic simplicity does neither justice.  (The Swedish tourists complaining about the noise of the "motorbikes" disturbing this "nature reserve" and lamenting, "I thought Sámi were supposed to be close to nature" represents such simplistic thinking.  Note that the Sámi at the end of the film are using helicopters and all-terrain vehicles to go mark the ears of the reindeer.  Elle-Marja does ask her sister to forgive her at the end, but that doesn't necessarily undo her years of changing her names, dress, and attitudes.  Unless, of course, we assume that she's simply awakening from years of "false consciousness": but isn't that as condescending as her teacher telling her that she can't become a teacher because her family needs her, she needs to use her skills on the land, and she's not smart enough to become a teacher?)  What is perhaps underemphasized in your argument is a sense of the complexities of postcolonial hybridity.  In other words, does Elle-Marja lose her status as "real Sámi" when she is in Uppsala and "southern Sweden"?  Is she only "real Sámi" when she is with the reindeer on the land?

Response: We agree and thank you for the detailed, concrete advice. We have expanded the discussion of the final scene to show its ambiguity in more detail, and added a few other examples which we hope clarify these points. Our impression is that the film presupposes that knowledge of the land and owning reindeer are still essential to Sámi identity. It is hard to show how the film does this without giving the impression we also think that (which we don’t). We hope the rephrasings have clarified that.

There are some seemingly powerful symbolic moments which could be analyzed and explicated more to good effect.  For example, 266 - what is the symbolic significance of Miss Holm behaving as though Elina is invisible?  For example, 383-84 - "She learns that her father was not local, and that her mother taught him about the fens."  For example, characters in both films holding their ears to the ground.  For example, the poem recited by Elle-Marja's teacher and her by Swedish-speaking Finnish poet Edith Sodergran, "I long for the land this is not."  Significance of names "Christina" and "Niklas"?  In Elina, the teacher repeatedly changing Anton's pen from his left hand to his right hand. Correct something with physical coercion.

Response: Due to the word restriction, we had removed some examples (the left hand example and the content of the lessons). However, we have developed those that we could include.

The character of Einar in Elina perhaps deserves some skeptical resistance.  Does he oversimplify the narrative (for children) by being the "good cop" to the female teacher's "bad cop"?  He wants to learn language from his students, goes out in solidarity with them, has eyes for the protagonist's mother, "rescues" Elina when her mother's apron isn't enough, drives off into the sunset with Elina and her mother at the end, giving them a ride from Elina's father's grave.  Is this "white knight" from southern Sweden a benign saviour for the northern women who can't quite make it on their own?

Response: We agree, but don’t feel we have space to develop the idea properly. We have added a more explicit comment on this point, but left it simply as a portrayal that might be easier for children to understand, but which is ahistorical.

Some specifics:

31 - "their ethnic minorities" rather than "its ethnic minorities" Corrected

36 - I believe the English title is Elina: As If I Wasn't There while the French title is L'Invisible Elina. Wikipedia uses the direct translation from Swedish (Elina: As If I Wasn't There). The Finnish version – Näkymatön Elina - literally translates as Invisible Elina and alludes to Tove Jansson’s short story. The Finnish DVD we used, had ‘Invisible Elina’ as the English title, but we have changed to the Wikipedia version.

63-64 - "a gap . . . was significant" vs. "were" Section removed

83 - "we do not intend to conflate the Sámi and the Tornedalingar" - Might you unpack this more by making a distinction between "indigenous" and other "national minorities"?
We have added a couple of sentences, but only mentioned points that are relevant for the film. The law only allows individuals to claim membership in one group (in terms of determining which language they can use with the authorities etc.), but as we point out, intermarriage and close proximity mean that that many people feel they belong to more than one group. Unpacking the distinctions beyond the films would easily result in stereotyping, especially in a short article.

110 ff. - here and later, you could make your "close reading" of this crucial part of the film more explicit (and hence more powerful): the violence of the flash of the photographer's camera; the specific measuring instruments (including the examining of the teeth inside the mouth--are reindeer measured in such ways?: horses are); the voyeurism (let's take a picture of the two of them holding hands); the enforced nudity (put your hands behind your heads); the complicity of the teachers).

Response: We have added more detail to the text.

153 - a mention of Canadian residential schools would add to this list (there are many overlaps, particularly with residential boarding schools which took children from their parents and communities with the intent of language re-education as well as industrial training)

Again – we did have this in the longer version and our larger project includes many such comparisons. We have now re-added it here as a small point.

176 ff. - Some specific quotations from the older Elle-Marja would make this come alive even more (She complains about "those people" lying, stealing, whining, their music is "shrill," "fucking circus animals," etc.)

Response: Done.

183-84 - This is crucial: it could be unpacked more. We have tried to do so.

217 ff. - I'm not convinced that "whilst the teenage Elle-Marja is easy to like and admire, the adult Elle-Marja in the frame story is deeply unlikeable."  Is not her "agency" as both adolescent and adult admirable?  It's worth noting that both the adolescent and the old woman pull a lock of hair over their wounded/marked ear.  Your argument seems to imply that Elle-Marja's whole adult life has been forced upon her and is redeemed only when she reconnects with the land, her language, her heritage.

Response: We have emphasized our point that in the 1930s a Sámi girl could not get an education. Elle-Marja chooses education – it is not forced upon her. We have also tried to tone down the idea of ‘likeability’ and emphasize moments of agency.

275 - "free will" vs. "freewill"

Response: Done.

313 - "trades sex for a place to spend the night": This seems a misrepresentation of what the film shows, namely, that EM seeks out Niklas at the dance, then goes to his home, then asks if she can stay, and then they have a consensual sexual relationship.  Even if he turns out unwilling or unable to stand up to his parents and friends, she has demonstrated her agency in initiating this relationship, has she not?

Response: The sentence has been rewritten.

333-34 - "Both films . . . suggest that agency within the Arctic landscape should determine who belongs there."  The implications of this claim would seem to open up "might makes right" and other unfortunate possibilities.  If only a minority of Sámi today live a traditional lifestyle on the land, does that make them the only "real Sámi"?

Response: The films are romantic and do suggest that connecting with the land is a way of being “true” to one’s ethnic identity. We have tried to clarify that this is the perspective evident in the films rather than our belief.  

442-45 - Is Elle-Marja's agency and resistance possible only through "knowing and acknowledging the land"?  Does she not demonstrate resistance both from her Swedish teachers and education and from her Sámi family and community?  Is she less Sámi in a library reading, reciting a Swedish poem, lying in an Uppsala park, changing her names and her clothing, staying in a hotel instead of a tent?  Is she "real Sámi" only when she is near reindeer?  How can one do justice to the complexity of her agency while still recognizing and exposing those things that were forced on her (including the enforced marking of her ear--which, as you note, may problematically be done by a fellow Sámi).  Does she not resist both her mother's "Speak Sámi" and her teacher's "Speak Swedish"?  Does she not live her life--as much as is possible for anyone--in a both-and rather than an either-or?

Response: We have tried to point out her “beside-ness”; she is both Sámi and Swedish, but society in the 1930s forces her to choose.

Finally, thanks for the article and for sharing two wonderful films with others!

Thank you for providing such helpful, constructive advice! It was much appreciated.

Submission Date

07 January 2019

Date of this review

12 Feb 2019 02:31:25